# Body Length and Craniometrics of Non-Native Raccoons in Two Regions in Middle Japan during Early Invasion Stages

**DOI:** 10.3390/ani13010055

**Published:** 2022-12-23

**Authors:** Takuya Kato, Fumiaki Yamasaki, Kandai Doi, Mieko Kawamichi, Shin-ichi Hayama

**Affiliations:** 1Faculty of Veterinary Science, Nippon Veterinary and Life Science University, Tokyo 180-8602, Japan; 2Japan Society for Promotion of Science Research Fellow, Forestry and Forest Products Research Institute, Ibaraki 305-8687, Japan; 3Kansai Wildlife Research Association, Kyoto 605-0981, Japan

**Keywords:** body length, cranial size, morphometrics, *Procyon lotor*, raccoon

## Abstract

**Simple Summary:**

Morphological characteristics related to sex-age class in introduced species during the early invasion stages, before further environmental adaptation, may indicate important information regarding their adaptability. For raccoons introduced to different regions in middle Japan, the results of multiple linear regression on body length showed that males were larger than females and body length increased with age class, while there was no difference between regions. The cranial size, which indicated different allometry in each sex, regressed positively according to increasing age class, body length, and body mass index in both sexes; however, it only differed between regions for females. Therefore, our results supported the hypothesis that older individuals with larger body sizes have an advantage in intrasexual selection and competition for food resources in raccoons. Considering the fact that multiple subspecies are sympatric in North America and the morphometrics of introduced raccoons in Japan were inconsistent with any of them, it is likely that hybridization occurred prior to introduction or naturalization. Further studies are necessary to demonstrate subspecific hybridization and its impact on morphological and ecological changes.

**Abstract:**

Although the genetic distribution of introduced raccoons (*Procyon lotor*) in recent years is well known, few studies have examined their morphometrics, especially the relationships between sex and age in the introduced populations. The aim of this study was to describe the morphological characteristics of raccoons from parts of eastern and western Japan during their early invasion stages, focusing on the relationships between body length and the principal component of craniometrics, with region, sex, age class, body length, and body mass index using a regression model. The body length increased more in males than females and in the older age class, supporting the association with intrasexual selection and competition for food resources. Positive relationships for body length and body mass index were found in craniometric analyses, particularly regarding cranial size components, in addition to age class for both sexes, while cranial size also differed between regions for females. The relationship between body length and craniometrics was inconsistent with that of subspecies originating in North America. Given the sympatric distribution of haplotypes of multiple subspecies without reproductive-isolating barriers in North America and in several introduced areas, hybridization must have occurred prior to the introduction or naturalization of this species.

## 1. Introduction

Globally, several carnivore species have been introduced to non-native areas, and they have managed to naturalize as invasive alien species owing to their high adaptability to various factors, including productivity and food availability, in each environment [1,2,3,4]. Following adaptation, the non-native population of a species may become ecologically and morphologically different from the population inhabiting the native region [5,6]. Compared to the indigenous animals, introduced raccoon dog (*Nyctereutes procyonoides*) populations in Europe demonstrate different craniometrics and non-metrical features [5,6]. Korablev and Szuma [6] proposed that the main factors influencing morphological variation in the raccoon dog could be related to the net primary production of ecosystems, adaptability to ecologically different regions, stochastic factors (such as the founder effect), and isolation of populations due to human activity. However, some omnivorous mammals seem to have rapid morphological changes in their new distribution [7,8,9,10]. Retrospective studies on the morphological characteristics during the early invasion stage in introduced species may provide insights regarding their adaptability in the future.

The Northern raccoon (*Procyon lotor*) is a medium-sized carnivore that is native to and widespread in North and Central America and has been introduced to Alaska [11], Russia [12], several countries in Europe [4,13], and Japan [1]. The raccoon has tolerance to a wide variety of bioclimatic conditions that results in its adaptation to various regions, and it is predicted that they will expand to new favorable areas north of the current favorable areas by 2050, due to climatic change [14]. This expansion may negatively impact the new areas because of the predation of native species of, for instance, amphibians, insects, and crustaceans [1,15]. The first raccoon naturalization in Japan was reported when 12 raccoons escaped from a zoo in Inuyama City, Aichi Prefecture in 1962 [16]; however, the raccoon had not yet been recognized as an invasive alien species. Miyashita [17] described the importation of over 20,000 raccoons to Japan as pets, following a famous television animation show in 1977. In Kamakura City, Kanagawa Prefecture, located in east central Japan, raccoon naturalization was confirmed in 1988 [18]; subsequently, the presence of naturalized raccoons was confirmed in most prefectures by 2006 [19]. The number of 5 km^2^ grids containing confirmed raccoon inhabitants increased from 1388 to 3862 between 2006 and 2018, which indicated an increase in distribution [20]. The growing invasive raccoon populations in Japan have become considerably problematic to the native ecosystem, agriculture, and public health [1].

Ecological characteristics, including productivity, home range, seasonal activity, feeding habits, and mortality, vary in different regions for North American raccoons, as described by Gehrt [21]. In addition, previous studies regarding the native raccoon have also clarified both sexual size dimorphism (SSD) and geographical variation in body size based on craniometrics. The SSD of the raccoon is known to involve larger body sizes for males than for females in each regional distribution [22,23]. Furthermore, the largest native population was observed in the northwestern region and the smallest in the southeastern region with a geographical gradient for intraspecific variation [24,25]. Geographic variation was also found in litter size, linked to the variation pattern of female raccoon body size [26]. Considering the widespread raccoon distribution in the northern and southern regions, development of ecological and morphological characteristics might be one of the most important factors explaining their adaptation to various environments.

Based on a study of mitochondrial DNA haplotypes of native raccoon distributions [27], introduced populations in the eastern part of middle Japan, including Chiba Prefecture [28] and Kanagawa Prefecture [29], are considered to originate from the central part of North America. Additionally, different haplotypes were found in several introduced populations in Japan [30,31], which suggests that independent introductions occurred and that the populations did not interact. Some reports on non-native raccoon populations demonstrated ecological characteristics regarding reproduction [32,33] and feeding habits [34,35,36] that suggest a high adaptability to new distribution areas. Although body size in some introduced carnivores may be related to their ecology [6,7,8], little is known about morphological characteristics, such as body length and craniometrics, for the introduced raccoons of Japan. The morphological characteristics of raccoons in the early stages of introduction are expected to provide important information for understanding its subsequent environmental adaptation as an invasive alien species.

The aim of this study was to describe the morphological characteristics of raccoons from parts of eastern and western Japan during their early invasion stages, focusing on the relationships between body length and the principal component of craniometrics with region, sex, age-class, body length, and body mass index using a regression model. We predict that: (1) SSD and age-related body length differences would occur even in non-native populations; (2) body length and craniometrics would differ between regions; and (3) the craniometrics and cranial growth pattern in each sex would be related to age class, body length, and nutritional condition which implicates an ecological background.

## 2. Materials and Methods

### 2.1. Study Site

The study was performed in Kamakura City, Kanagawa Prefecture and Maizuru City, Kyoto Prefecture; these sites were chosen to represent the eastern and western parts of middle Japan, and the distance between the two areas is approximately 500 km (Appendix A). Although Kamakura is located on the Pacific side and Maizuru on the side of the Sea of Japan, both regions have temperate humid climates. The mean annual temperatures and annual precipitation in Kamakura and Maizuru are 16.2 °C and 1730.8 mm and 14.8 °C and 1941.2 mm, respectively, which indicate that there is little difference between the climates of the two regions.

### 2.2. Samples and Measurement

Raccoon carcasses were collected from 2005 to 2007 in Kamakura City, and from 2009 to 2014 in Maizuru City. Considering the timing of first reports and the progress of the invasion in Kamakura and Maizuru, each raccoon collection period in this study can be assumed as the early post-introduction stage. All raccoons were captured by trappers permitted by municipalities using box traps. The raccoons were then euthanized by CO_2_ inhalation according to the Guidelines for the Management of Invasive Alien Species in Japan [37].

Postmortem examination in each carcass identified the gender by genital observation, and tooth eruption was assessed to exclude individuals under five months of age when pre-weaned [38,39]. Body weight (in g) and length (in mm: nose to tail, subtracting the tail length) were then measured. To analyze the body condition and nutritional status of individuals, the body mass index (BMI) was estimated using the formula W/L^2^, where W is the body weight expressed in kg and L is the body length expressed in meters [40].

The skinned head was removed, and the skull was processed to estimate age class and for further craniometrics. Based on the size of the canine root apical foramen [41] and cranial suture obliteration [42], all individuals were classified into three age groups: 5–11 months of age, juveniles; 12–23 months of age, yearlings; and ≥24 months of age, adults. Consequently, the number of raccoon samples were 143 from Kamakura (15 juveniles, 30 yearlings, and 15 adult females; 16 juveniles, 34 yearlings, and 33 adult males) and 63 from Maizuru (6 juveniles, 10 yearlings, and 10 adult females; 11 juveniles, 16 yearlings, and 10 adult males). Measurements were performed for 13 cranial components using digital calipers with 0.01 mm precision as per previous studies [24,25,43,44] (Figure 1). The dataset of all specimens analyzed in this study is shown in Appendix A.

### 2.3. Data Analysis

#### 2.3.1. Body Length by Region, Sex, and Age Class

The correlation of body length with sex, age class, and region was tested using a multiple regression analysis. The body length was checked for normality and homogeneity, set as the outcome variable without transformation of raw data, whereafter the region, sex, and age class were entered as explanatory variables, and was examined for interactions between sex-age classes. Multicollinearity between the explanatory variables was assessed using the variance inflation factor (VIF) [45]. The best model selected using the Akaike information criteria (AIC) as well as other informative models with ΔAIC < 2 [46] were evaluated in terms of coefficient parameters and significance of explanatory variables.

#### 2.3.2. Craniometrics in Each Sex

For raccoons, sexual dimorphism was reported for craniometrics in previous studies [23,43]; therefore, further analyses were conducted separately for female and male raccoons. Since it is possible to summarize the number of variables to a small number of composite data, principal component analysis (PCA) is appropriate for craniometric studies [25,47,48]. In order to derive principal components (PCs), the PCAs for each sex were performed using the correlation matrix on the raw data, consisting of 13 variables of cranial measurements, without log transformation as described by O’Keefe et al. [46].

#### 2.3.3. Relationships between Cranial Size and Basal Factors

To evaluate the basal factors related to craniometrics, significant PCs were considered as summarized cranial characteristics for each sex using the eigenvalue and screen plot, and the relationship between the PCs and basal factors was analyzed using a multiple regression model [47,49]. The basal factors included region, age class, body length, and BMI and were entered as explanatory variables, which confirmed multicollinearity based on the VIF among each other [45], and the multiple regression analyses in each sex were performed for the PCs as outcome variables. The best model and other valid models with ΔAIC < 2 were selected using the Akaike information criteria (AIC) to examine factors related to the cranial characteristics in each sex [46].

#### 2.3.4. Allometry of Craniometrics in Each Sex

To determine cranial growth patterns of raccoon in each sex, we used a multivariate approach based on the generalized allometry equation proposed by Jolicoeur [50]. In this theory, a global multivariate allometry vector uses the first PC value (PC1) calculated from PCA for a variance-covariance matrix of log-transformed variables; if it derives a strong size-factor that obtained high eigenvalue, measurements are positive. Confidence intervals (CI) for each coefficient were estimated from sampling distributions based on 1000 bootstrap iterations. The isometric vector in this study was 0.277 derived by 1/√p, with p representing the number of variables [50].

#### 2.3.5. Functions for Statistical Analysis

Statistical analyses were performed using R. ver. 4.1.2 [51], using the functions *VIF* in the package “car” for VIF checking [52], and *lm* and *princomp* in the package “Stats” for the regression model and PCA [51], respectively; the function *dredge* in the package “MuMIn” was used for valid model selection using the AIC [53]; and the function *boot* in the package “boot” to calculate the 95% CI of the global allometry vector [54].

## 3. Results

### 3.1. Body Length by Region, Sex, and Age Class

For all 206 specimens, the descriptive analysis showed that body length differed between sexes within each age class and region. For adult females and males, the mean ± SD of body lengths was 557.0 ± 28.3 and 569.4 ± 35.6 in Kamakura and 540.4 ± 30.0 and 592.7 ± 28.1 in Maizuru, respectively (Appendix A). The violin plots of body length (mean ± SD) per region, sex, and age class showed that males were larger than females in each age class and had increased body lengths with older age class. Moreover, despite the high variance of yearling males, little difference was found between regions (Appendix A). As a result of the regression analysis for body length, the region was excluded from the best model (model 1; Adjusted R^2^ = 0.35, *F*_(3, 303)_ = 37.39, *p* < 0.001: Table 1), while the body length was significantly correlated with both sex (*p* < 0.001) and age classes (*p* < 0.001). In the best model, male raccoons were larger than the females (coefficient = 31.83, 95% CI = 22.52–41.14) and juveniles were smaller than yearlings (coefficient = 30.24, 95% CI = 18.49–41.99) and adults (coefficient = 49.12, 95% CI = 36.71–61.53). Although, in other valid models with ΔAIC < 2, sex-age interaction (model 2) and region (model 3) were selected in addition to sex and age class, these were not significantly correlated to the body length despite indicating sex and age class strongly affected the body length.

### 3.2. Craniometrics in Each Sex

Regarding craniometrics, the mean values of males were higher than those of females of all age classes and regions, except for the length of the maxillary tooth row, greatest width across upper molars (M1s), greatest width across lower third premolars (P3s), rostral width, width of incisors, and least distance between lower fourth premolars (P4s) (Appendix A). In a PCA using craniometrics, the axes of PC1, PC2, and PC3 indicated 56.79%, 12.73%, and 7.03% for females and 58.19%, 9.95%, and 6.89% for males, respectively (Table 2). The positive and high coefficients of all measurements for PC1 in both sexes indicated large size variations, indicating that these were defined as robust size factors. Contrary to those of PC1, the coefficients of PC2 were a mixture of positive and negative coefficients for both females and males, indicating that there is a shape factor. The individual scores for the PC1 and PC2 axes by region and age class in each sex are plotted in Figure 2, showing that craniometric size varied with age class and region in females, but only with age class in males.

### 3.3. Relationships between Cranial Size and Basal Factors

Following O’Keefe et al. [46], because of their positive and high coefficients, the PC1s of both sexes were used as size factors in craniometrics; therefore, region, age class, body length, and BMI were used in the regression analysis (Table 3). For the best model in females (Adjusted R^2^ = 0.68, *F*_(5, 80)_ = 36.68, *p* < 0.001: Table 3), PC1 indicated that the cranial size of raccoons in the Maizuru region was significantly smaller than that in the Kamakura region (coefficient = −2.09, 95% CI = −2.85–−1.33) and for juveniles were smaller than for adults (coefficient = 2.03, 95% CI = 0.94–3.11). The PC1 value of females was also positively correlated with body length (BL) and BMI (BL: coefficient = 0.04, 95% CI = 0.02–0.05; BMI: coefficient = 0.14, 95% CI = 0.01–0.26: Table 3). There were no other valid models with ΔAIC < 2 for the PC1 in females. On the other hand, the PC1 value of males in the best model (Adjusted R^2^ = 0.54, *F*_(4, 115)_ = 35.71, *p* < 0.001: Table 3) indicated that juveniles were smaller than yearlings (coefficient = 1.30, 95% CI = 0.39–2.31) and adults (coefficient = 1.76, 95% CI = 0.69–2.84). In addition, the PC1 value of males was also positively correlated with BL and BMI (BL: coefficient = 0.03, 95% CI = 0.02–0.04; BMI: coefficient = 0.37, 95% CI = 0.25–0.50) in the best model (Table 3). However, the PC2 value of females was slightly positively correlated with only BL in females for all valid models including the best model (Adjusted R^2^ = 0.12, *F*_(4, 81)_ = 3.94, *p* < 0.01; BL: coefficient = 0.02, 95% CI = 0.01–0.02; Table 3), whereas the PC2 value of males was high in Maizuru (region: coefficient = 0.85, 95% CI = 0.45–1.25) and positively slight correlated with BL (coefficient = 0.01, 95% CI = 0.004–0.01; best model: Adjusted R^2^ = 0.19, *F*_(2, 117)_ = 15.10, *p* < 0.001; Table 3).

### 3.4. Allometry of Cranial Size in Each Sex

The global allometry vector presented by the bootstrap allometric coefficient and 95% CI of cranial size differed in positive or negative patterns against isometry between sexes (Figure 3, Appendix A). Variables changing with significant negative allometry included greatest length of skull, basal length, and length of maxillary toothrow; however, positive allometry was only represented by zygomatic width, which indicated that it is associated with sex. Although the cranial width in females and greatest width across first upper molars and greatest width across lower third premolars in male were found to have significant negative allometry, positive allometry was found for paraoccipital width and least distance between lower fourth premolars in males.

## 4. Discussion

Body length and craniometrics of mammalian species are standard measurements used to understand their morphological characteristics, including those used to test Biogeography theories and identify alien species [6,55]. For native raccoon populations, although the cranial morphology is well-understood [24], only a few reports describe systematic body length details. Some studies have reported the body lengths of introduced raccoons in Germany [43], Spain [44], and Japan [56]; however, it is unclear how the body length differs with sex and age classes. This is the first study to report that the body length and craniometrics of introduced raccoon populations during their early invasion stage are related to sex and age classes (Table 1 and Table 3).

The evolution of SSD has been explained by three major hypotheses based on: sexual selection, intersexual food competition, and reproductive role division [57]. In raccoons, SSD has been associated with body length, namely in terms of the larger size of males compared to that of females [22,58]. Since raccoons are typically promiscuous [21], their SSD may be caused by sexual selection. However, if body size increases with age in not only males but also females, resulting in an advantage for intrasexual selection or food competition, the SSD of raccoons could indicate a relatively low value compared to differences in age class. While older males with large body size have been considered to dominate over smaller individuals, larger individuals (regardless of sex) have an advantage, particularly in the competition for food resources [58,59]. Isaac [60] suggested that the pattern of reproductive efforts in the common brushtail possums (*Trichosurus vulpecula*) differed with regard to age in males, and the older, more dominant individuals may invest less time than younger males in finding receptive females owing to a hierarchal social system, resulting in an increase in foraging behavior. In addition, female reproductive characteristics of raccoons, including pregnancy rates and litter sizes, were increased for adults compared to yearlings in both native and introduced areas [21,32,33,61]. Our study demonstrated that both sex and age classes were positively related to the body length (Table 1); the resulting differences in body length by the sex-age class of raccoons indicated an association with intrasexual selection for both sexes and competition for food resources.

Regarding cranial characteristics, the SSD of raccoons in native distributions revealed that males were larger than females [23]. The PC1s for both sexes derived for introduced raccoons in Kamakura and Maizuru could be considered cranial size indicators as described by O’Keefe et al. [47]. Our study demonstrated that the PC1s were high for the older age class and had a week positive correlation with body length and BMI (Table 3), except between juveniles and yearlings in females. In the allometry assessment using PC1 as a size indicator, the zygomatic width of both sexes and the paraoccipital width correlations of males were found to be positive (Figure 3), which are important for masseter muscle attachment. Given that in raccoons, juvenile male dispersal is known to occur earlier than female dispersal [62], the difference between sexes in cranial growth pattern can be considered an important characteristic. Consequently, our results on cranial size also suggest that older individuals exercising dominance in the social group have larger body sizes [59]. In addition, the BMI of female raccoons in Kamakura was reported to increase with their growth until maturity despite seasonal fluctuations [63]. Based on BMI evaluation of the body condition index of raccoons [40,64], individuals with high BMI can be considered to have good physiques. Raccoon cranial width dimensions in particular, indicated an advantageous factor for competition among individuals and was related to body length growth and BMI in the two introduced populations in middle Japan.

The regression model for body length indicated no difference between Kamakura and Maizuru raccoon populations (Table 1). However, slight differences were found in the craniometrics between regions in the sizes (PC1) of females and the shapes (PC2) of males (Table 3). Goldman [22] described that at least three subspecies including *P. lotor lotor*, *P. l. hirtus*, and *P. l. varius* have widespread distribution in North America, and that the body length increases in the order of *P. l. varius* < *P. l. lotor* < *P. l. hirtus* based on the measurement of few specimens of each subspecies in several regions. The body lengths of both sexes of adult raccoons in Kamakura and Maizuru tended to be close to those of *P. l. lotor* in Virginia [22], as shown by the least square mean analysis (females: *F*_(1, 1373)_ = 1.64, *p* = 0.22; males: *F*_(1, 95)_ = 0.08, *p* = 0.78). However, craniometrics for individuals from both regions indicated a similar size compared to *P. l. hirtus*, which was a larger subspecies than *P. l. lotor* [22,24]. Hence, the introduced raccoons in Kamakura and Maizuru were found to be morphologically inconsistent with subspecies in North America.

Genetic structure evaluation of raccoons has indicated an absence of reproductive-isolating barriers due to the degree of lineage mixing across subspecific native distributions [27]. Some of the haplotypes across the subspecies distribution in native regions have been found in raccoons introduced into Europe and Japan [65,66]; this has led to the hypothesis that raccoon populations had already undergone subspecific hybridization before their distribution and introduction into these regions. The raccoons introduced into Kamakura and Maizuru seem to be independent in terms of their genetic background [30,31]; which we consider to be an important reason for the slight morphological differences in body size between these populations. The first raccoon naturalizations in Kamakura City and Maizuru City were reported in 1988 [18] and 2001 [67], respectively. Capture removal of raccoons has subsequently been promoted since 2006 in Kamakura and 2009 in Maizuru by the Raccoon Control Program, which was formulated by the municipalities based on the Invasive Alien Species Act 2005. Considering that raccoon naturalization occurred in the 1980s [18], mating occurs once yearly, and females mature at only 1 year of age [33], this species has not yet been considered as affected by environmental factors because there have been only 20–30 generations since its establishment. Comparisons between morphological variations in introduced and indigenous populations have been reported in some carnivore species. These include the raccoon dog [6], American mink (*Neovison vison*) [7,10], and the small Indian mongoose (*Herpestes javanicus*) [8], in which the size variability of the introduced populations is connected with the invasion processes as well as the history of the introduction site, in terms of factors such as the animal fur industry, adoption as pets, the number of naturalized animals, and the influence of several biotic and abiotic environmental factors.

## 5. Conclusions

Non-native raccoons introduced to Kamakura and Maizuru present that: (1) SSD and that individuals of the older age class have larger body length compared to the other age classes; (2) no major differences of body length and craniometrics were found between regions, indicating that the distance is not influence their morphologies; (3) the craniometrics and cranial growth pattern for the cranial width dimensions in particular, related to age class, body length, and body mass index implicate an important factor for competition among individuals in each sex. These characteristics may be related to the ecology of the raccoons, including intrasexual selection for both sexes and competition for food resources. Future studies should investigate the morphological differences among raccoon populations in various introduced areas and the morphological changes of the introduced populations over the next few decades, focusing on factors such as genetic differentiation, environmental adaptation, and population density.

## Figures and Tables

**Figure 1 animals-13-00055-f001:**
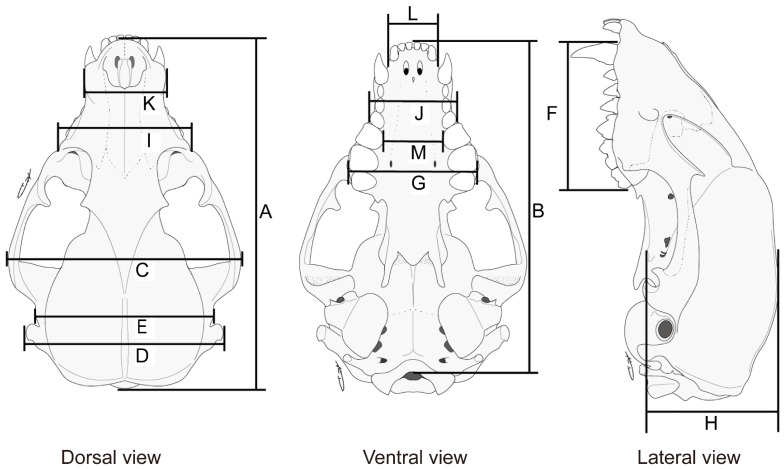
Lateral, ventral, and dorsal craniometrics of the raccoon (*Procyon lotor*) skull. A: greatest length of skull (GLS), B: basal length (BaL), C: zygomatic width (ZW), D: paraoccipital width (PW), E: cranial width (CW), F: length of maxillary tooth row (LMT), G: greatest width across first upper molars (M1s) (GWM), H: depth of skull (DS), I: least width between infraorbital foramen (LWIF), J: greatest width across lower third premolars (P3s) (GWP), K: rostral width (RW), L: width of incisors (WI), M: least distance between lower fourth premolars (P4s) (LDP).

**Figure 2 animals-13-00055-f002:**
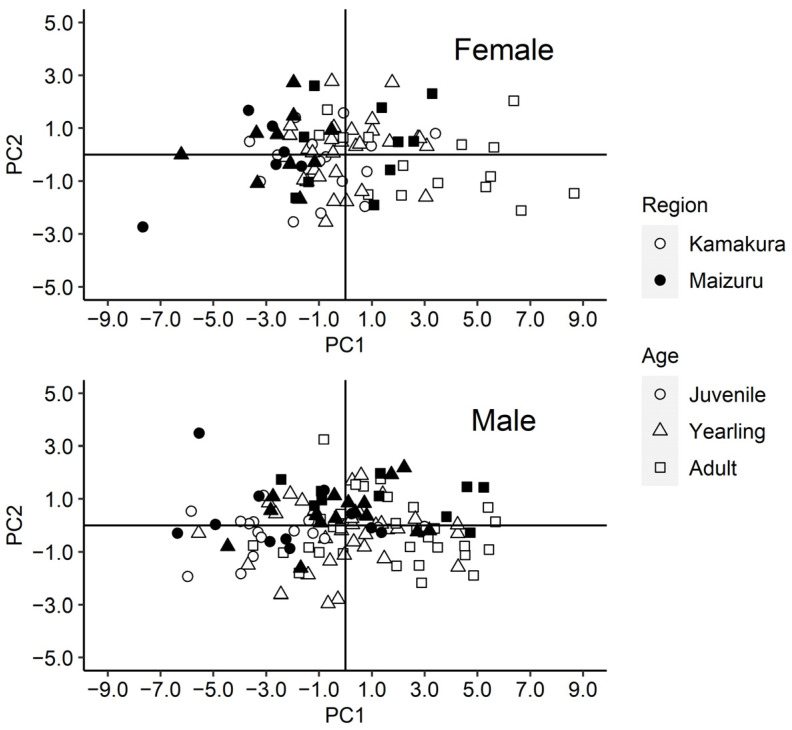
Principal component (PC) analysis of skull measurements in each sex of introduced raccoons in Kamakura and Maizuru, Japan. The white and black colors indicate Kamakura and Maizuru, respectively; and the shape indicate that circle, triangle, and cube are juvenile, yearling, and adult, respectively.

**Figure 3 animals-13-00055-f003:**
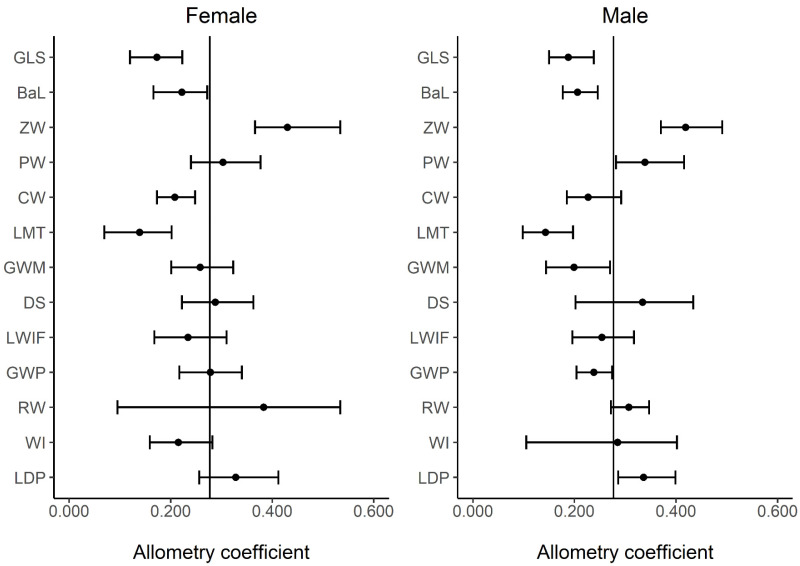
Multivariate allometry vector (the 1st eigenvector derived from the covariance matrix of ln-transformed variables) and 95% confidence interval using the bootstrap resampling procedure for skull morphometrics in each sex of introduced raccoons in Kamakura and Maizuru, Japan. The isometric value is 0.277 according to Jolicoeur [50]. Measurement letters corresponded to those of Figure 1. GLS, greatest length of the skull; BaL, basal length; ZW, zygomatic width; PW, paraoccipital width; CW, cranial width; LMT, length of maxillary tooth row; GWM, greatest width across first upper molars (M1s); DS, depth of skull; LWIF, least width between infraorbital foramen; GWP, greatest width across lower third premolars (P3s); RW, rostral width; WI, width of incisors; LDP, least distance between lower fourth premolars (P4s).

**Table 1 animals-13-00055-t001:** Multiple linear regression analysis for body length with region, sex, and age class of feral raccoons in Japan. The best models were selected using the Akaike information criteria (AIC) and valid models within ΔAIC < 2.0.

Category	Variable	Regression Coefficient ^1^	Standard Error	95% CI ^2^	VIF
Model 1 (Best model): Adjusted R^2^ = 0.35, *F*_(3, 202)_ = 37.39, *p* < 0.001, AIC = 2035.4					
Sex	Female	Reference			1.00
	Male	31.83 ^†^	4.72	22.52–41.14	
Age class	Juvenile	Reference			1.00
	Yearling	30.24 ^†^	5.50	18.49–41.99	
	Adult	49.12 ^†^	5.96	36.71–61.53	
(Intercept)		496.57 ^†^	5.79	485.73–507.41	
Model 2: Adjusted R^2^ = 0.35, *F*_(5, 200)_ = 23.26, *p* < 0.001, AIC = 2035.9					
Sex	Female	Reference			2.06
	Male	23.62 *	9.67	4.56–42.69	
Age class	Juvenile	Reference			1.55
	Yearling	20.28 *	8.95	2.63–37.94	
	Adult	49.17 ^†^	9.84	29.77–68.56	
Interaction	Sex–Yearling	17.82	11.96	−5.77–41.41	1.94
	Sex–Adult	0.83	12.78	−24.37–26.03	
(Intercept)		501.19 ^†^	7.25	486.89–515.48	
Model 3: Adjusted R^2^ = 0.34, *F*_(4, 201)_ = 27.91, *p* < 0.001, AIC = 2037.3					
Region	Kamakura	Reference			1.00
	Maizuru	0.77	5.06	−9.22–10.75	
Sex	Female	Reference			1.00
	Male	31.83 *	4.73	22.49–41.16	
Age class	Juvenile	Reference			1.00
	Yearling	30.29 *	5.98	18.50–42.09	
	Adult	49.17 ^†^	6.32	36.71–61.62	
(Intercept)		496.31 ^†^	5.79	484.89–507.72	

^1^ *p*-values refer to Ho: Regression coefficient, 0; *: *p* < 0.05; ^†^: *p* < 0.001. ^2^ 95% confidence interval (CI) of the regression coefficient.

**Table 2 animals-13-00055-t002:** Factor loadings of principal component analysis (PCA) in cranial measurements in each principal component (PC) for each sex of introduced raccoons in Kamakura and Maizuru, Japan.

	Female			Male		
	PC1	PC2	PC3	PC1	PC2	PC3
Eigenvalue	7.38	1.65	0.91	7.56	1.29	0.90
Proportion	56.79	12.73	7.03	58.19	9.95	6.89
Cumulative	56.79	69.52	76.55	58.19	68.14	75.03
Measurement loadings ^1^						
GLS	0.28	0.44	0.03	0.30	0.36	0.06
BaL	0.30	0.36	0.01	0.31	0.22	0.11
ZW	0.32	−0.07	−0.30	0.32	−0.02	−0.20
PW	0.30	0.14	−0.40	0.30	0.21	−0.11
CW	0.30	0.13	−0.35	0.30	0.28	0.03
LMT	0.20	0.38	0.53	0.23	0.29	0.19
GWM	0.27	−0.40	0.01	0.22	−0.46	0.43
DS	0.30	−0.01	−0.17	0.26	−0.14	−0.51
LWIF	0.26	−0.13	0.41	0.28	0.08	0.29
GWP	0.28	−0.38	0.23	0.27	−0.37	0.14
RW	0.27	−0.01	0.13	0.30	−0.01	−0.25
WI	0.24	−0.07	0.27	0.21	−0.36	−0.45
LDP	0.26	−0.39	−0.08	0.27	−0.34	0.29

^1^ Measurement letters corresponded to those of Figure 1. GLS, greatest length of skull. BaL, basal length. ZW, zygomatic width; PW, paraoccipital width; CW, cranial width; LMT, length of maxillary tooth row; GWM, greatest width across first upper molars (M1s); DS, depth of skull; LWIF, least width between infraorbital foramen; GWP, greatest width across lower third premolars (P3s); RW, rostral width; WI, width of incisors; LDP, least distance between lower fourth premolars (P4s).

**Table 3 animals-13-00055-t003:** Multiple regression analysis for principal component (PC) scores regressed against region, age class, body length (BL), and body mass index (BMI) of female and male introduced raccoons in Japan.

		Regression Coefficient in Modeled Variables						
		Region ^1^	Age Class ^3^		BL	BMI	AIC	ΔAIC
			Yearling	Adult				
PC1	Female							
	model 1 (Best model)	−2.09 ^†^	0.13	2.03 ^†^	0.04 ^†^	0.14 *	327.5	0.00
	Other valid models with ΔAIC: none							
	Male							
	model 1 (Best model)		1.35 *	1.76 *	0.03 ^†^	0.37 ^†^	498.5	0.00
	model 2	−0.39	1.30 *	1.66 *	0.03 *	0.38 *	499.3	0.86
PC2	Female							
	model 1 (Best model)	0.49	0.09	−0.60	0.02 ^†^		284.1	0.00
	model 2				0.01 *		284.2	0.15
	model 3	0.38			0.01 *		284.5	0.39
	model 4				0.01 *	−0.06	284.6	0.55
	model 5		0.10	−0.50	0.01 *		285.1	0.97
	model 6	0.32			0.01 *	−0.05	285.4	1.34
	model 7	0.44	0.08	−0.58	0.02 ^†^	−0.03	285.6	1.46
	model 8		0.07	−0.48	0.02 ^†^	−0.05	285.9	1.77
	Male							
	model 1 (Best model)	0.85 ^†^			0.01 ^†^		352.0	0.00
	model 2	0.85 ^†^			0.01 ^†^	0.02	353.6	1.63

^1^ *p*-values refer to Ho: regression coefficient, 0. *: *p* < 0.05. ^†^: *p* < 0.001. ^3^ The variable is categorical, which means “Yearling” and “Adult” coefficients against “Juvenile (Reference)”, respectively.

## Data Availability

The data presented in this study are available in Appendix A.

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
