# Peer review of "Body Length and Craniometrics of Non-Native Raccoons in Two Regions in Middle Japan during Early Invasion Stages"

_animals, 2022, doi:10.3390/ani13010055_

Round 1
Reviewer 1 Report
Congratulations on the study, it shows some interest in regions of the world with this species of animals and close family, but needs to be restructured, has confusing parts. Material and methods in the introduction, results/discussion appear in the Conclusion, need to make a profound change in the work , talk about all the tables and figures that are presented in the Results, and Discuss about the topic and not about assumptions, and a short Conclusion and about what resulted from the study. English not being my mother tongue, I won't go into details, but it needs to be improved, several expressions are translated to the letter and must be corrected. Below are some details to improve.
It is important to specify and say what the Purpose of the study is... the Abstract can start with a small introduction, then clear objectives, until you start the sentence with, “The aim…”, you can talk a little about the material/methods or go later to results and End with a concluding sentence.
Line 27: You can start by replacing “We analyzed” with “The aim of the study was…”
Line 46 – 47: Insert a reference to this statement.
Line 99: Swap “Therefore, the purpose…” for “The aim…” and improve the description of the Purpose of the study they present… you can put at the end of the Introduction, the same sentence of the Purpose that you put in the Abstract, it can and should be the same.
Lines 101 – 106: Delete these phrases are here too, that would be material and methods.
Lines 201-202: It is not good to indicate the results, telling them to go and see table X or Y, talk a little about the results and then put them in parentheses (Table X or Y).
Lines 205-206: the same as indicated for lines 201-202.
Lines 213-215: Here the same, they write the caption of Figure S1 and nothing else… you can't. If the best Model was 1, why do they present Model 3.
Lines 237-238: Same as saying, they need to talk a little about the results and put them between relatives (Table S3), they can't be constantly telling readers to go see the tables.
Table S3: What are the numbers in small, below the larger numbers in Bold… They need to indicate SD.
Lines 243-245: the same as indicated for lines 201-202.
Figure 2: What are the black squares and triangles? Need to indicate in Figure.
Lines 264-264: Material and methods.
The values ​​they present must be interpreted with care, since the models lack consistency by the R2 values ​​(expresses the amount of data variance that is explained by the linear model).
Line 296: They don't need this sentence, they just put it between relatives below when they talk about the results.
Figure 3: The graphs do not show the same x-axis values ​​(it is very close) but they can be the same.
Line 324: “To the best of our knowledge, this is the first study” it is important to be sure.
Line 349: “defined as robust size indicators” the number of animals is not high, the model not indicated was not high… be careful when indicating so much certainty.
Be careful when using words like: Slightly, small/subtle differences, deduce
Lines 371-374: You have to talk about your study and not the studies of others and if you didn't do it in yours, then don't talk, but if you want to compare, do basic statistical analysis using LSmeans.
Lines 391-392: This is a baseless assumption, you shouldn't make assumptions, just talk about what you show in your results.
Conclusions:
Lines 397-402: This is not a conclusion. They need to rewrite and reduce, starting by eliminating these sentences.
Author Response
Dear Reviewer 1,
We wish to express our appreciation to the reviewers for their insightful comments on our paper. The comments have helped us to significantly improve the paper.
We hope that our edits and the responses satisfactorily address all the issues and concerns you have noted.
Again, thank you for giving us the opportunity to strengthen our manuscript with your valuable comments and queries.
Sincerely,
Takuya Kato

Reviewer 2 Report
The MS is written in a good English language.
1. Title. I would not say “feral”. Better “alien” or “non-native”. Feral mostly refers to domestic species, and raccoons are not domestic.
2. Line 58. Raccoon should be “northern raccoon” at least the first time it is mentioned.
3. Line 59. “introduced to”
4. Lines 99-106. Please, report your predictions. Also, in the Introduction, remember that the raccoon is effectively an invasive species with several impacts shown throughout its invasive range. See for intance: Boncompagni L., Molfini M., Ciampelli P., Fazzi P., Lucchesi M., Mori E., Petralia L., Mazza G. (2021). No country for native crayfish: importance of crustaceans in the diet of native and alien Northern raccoons. Ethology, Ecology and Evolution 33: 576-590.
5. Methods and results are clear, well-cited and replicable.
6. Lines 319 – 326. Cranial measurements are widely reported to be a useful method to identify alien species or sexual-size dimorphism. For instance check this paper: Mori E., Ancillotto L., Lovari S., Russo D., Nerva L., Mohamed W.F., Motro Y., Di Bari P., Plebani M. (2019). Skull shape and Bergmann’s rule in mammals: hints from Old World porcupines. Journal of Zoology 308: 47-55.
Author Response
Dear Reviewer 2,
We wish to express our appreciation to the reviewers for their insightful comments on our paper. The comments have helped us to significantly improve the paper.
We hope that our edits and the responses satisfactorily address all the issues and concerns you have noted.
Again, thank you for giving us the opportunity to strengthen our manuscript with your valuable comments and queries.
Sincerely,
Takuya Kato

Round 2
Reviewer 1 Report
Dear Authors,
The review carried out followed what was requested, with the exception of the conclusion that can be improved.
You put the detailed "aim" at the end of the introduction well, so the conclusion should reflect the 3 points you refer to as objectives.
I propose that you remove the beginning: "In conclusion, this study showed that" you don't need to write conclusion or study in the conclusion, you can start right away: "Non-native raccoons introduced..."
When you say: "Apart from the small difference in craniometric values between regions, no major morphological difference was found between the non-native raccoon populations from Kamakura and Maizuru." must conclude and not talk about the results, they can say that the distance (between regions) did not influence the morphology.
I have a question regarding the last line of the conclusion: New studies should study the newly introduced non-native raccoon... or study the animals introduced (relatives of the animals that evaluated) with a greater number of generations, as they refer to in the work?
Congratulations on the work and thank you for your response.
Author Response
Dear Reviewer 1,
We wish to express our appreciation to the reviewer for its insightful comments on our paper. The comments have helped us to improve more strongly the paper.
We hope that our edits and the responses satisfactorily address all the recommendation and concerns you have noted.
Thank you again for giving us the opportunity to strengthen our manuscript with your valuable comments and queries.
Sincerely,
Takuya Kato
